

# Identifying quantum phase transitions with minimal prior knowledge by unsupervised learning

Mohamad Ali Marashli[1], Ho Lai Henry Lam[1], Hamam Mokayed[2],
Fredrik Sandin[2], Marcus Liwicki[2], Ho-Kin Tang[3] and Wing Chi Yu[1]⋆

**1** Department of Physics, City University of Hong Kong, Kowloon, Hong Kong
**2** Department of Computer Science, Electrical and Space Engineering,
Luleå University of Technology, 971 87 Luleå, Sweden
**3** School of Science, Harbin Institute of Technology, Shenzhen, 518055, China

⋆ wingcyu@cityu.edu.hk

## Abstract

In this work, we proposed a novel approach for identifying quantum phase transitions in one-dimensional quantum many-body systems using AutoEncoder (AE), an unsupervised machine learning technique, with minimal prior knowledge. The training of the AEs is done with reduced density matrix (RDM) data obtained by Exact Diagonalization (ED) across the entire range of the driving parameter and thus no prior knowledge of the phase diagram is required. With this method, we successfully detect the phase transitions in a wide range of models with multiple phase transitions of different types, including the topological and the Berezinskii-Kosterlitz-Thouless transitions by tracking the changes in the reconstruction loss of the AE. The learned representation of the AE is used to characterize the physical phenomena underlying different quantum phases. Our methodology demonstrates a new approach to studying quantum phase transitions with minimal knowledge, small amount of needed data, and produces compressed representations of the quantum states.

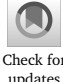

# 1  Introduction

Understanding the quantum phases and phase transitions of quantum many-body systems is a fundamental problem in condensed matter physics. Different phases give rise to physical phenomena such as superconductivity and topological insulators [1] which can have a wide range of applications [2,3]. Detecting and characterizing these transitions in quantum many-body systems is often challenging and requires extensive study of the systems or significant computational resources.

Traditional methods rely on the knowledge of the order parameters to detect phase transitions in quantum systems [4]. These order parameters serve as indicators of the system's state and its transitions between different phases. However, finding a suitable order parameter is a highly non-trivial task, especially in topological systems. In topological systems, the order parameter is usually non-local, meaning it cannot be described by local observables at a single point in the system. Instead, it often involves correlations between distant parts of the system, making its identification and measurement more challenging. Recent advancements in the study of topological phases have highlighted the importance of non-local string order parameters in capturing the unique properties of these systems [5].

Other popular approaches of detecting phase transitions involve measuring the entanglement [6–8] or the correlation length in the infinite Density Matrix Renormalization Group (iDMRG) [9]. These methods do not define an order parameter for the phase transition but attempt to provide information about how correlation changes in the system which often corresponds to a phase change. However, they can be model or phase specific and may not always work. The entanglement is not uniquely defined and there exists ambiguity in partitioning the system so that the entanglement measured can signal the transitions between different phases. The correlation length in iDMRG also does not always possess significant changes across phase transitions (see an example in Appendix A). Furthermore, the use of iDMRG requires translational symmetry and an area-law entanglement, which can limit its application in, for example, disorder systems or states of matters that are determined by long-time dynamics.

With machine learning (ML) techniques being developed to analyze large data systems, recent studies have shown they can be efficient tools for solving problems in natural sciences [10] such as biology [11,12],chemistry [13] and physics [14] including identifying and characterizing quantum phases and phase transitions [15–19]. Early works such as Ref. [20] and Ref. [21] used supervised learning with binary classifier neural network to detect phase transition in the Ising model, and many-body localization mobility edge in the spin-1/2 Heisenberg chain in a random external field, respectively. These works demonstrated the viability of neural net-

works in detecting phase transitions in equilibrium and out-of-equilibrium systems. However, they had the limitations of needing labeled data, thus prior knowledge of the phase diagram, for supervised training and was only demonstrated for binary classification of a single transition.

Since then new unsupervised machine learning techniques have been introduced to detect phase transitions in a variety of models without the need of labeled data and without empirical knowledge of the order parameters [17–19]. Examples of promising recent works in this field are Chung *et al.* which used spin-spin correlation as input with Autoencoders (AE) and K-means clustering to find the transition points [17], and Han *et al.* which used Monte Carlo state configurations as input for an unsupervised contrastive learning inspired by SimCLR architecture [22] to find the phase transitions [18]. However, these works have some drawbacks in limitation and requirements, for example in Chung *et al.* work the choice of the spin-spin correlation functions for each system affects the results [17], making prior knowledge and understanding of the system essential for accurately determining the transition points. While in Han *et al.* work, up to $10^5$ state configurations are required for each state in the driving parameter space [18], thus demanding significant computational power and limiting the ability of exploring systems with multiple driving parameters.

On the other hand, Kottmann *et al.* used the entanglement spectrum as input and traced the loss of an AE with symmetric connections trained on a single phase to obtain the phase diagram of the one-dimensional (1D) extended Bose-Hubbard model [19]. The working principle is similar to the fidelity approach which measures the similarity between two quantum states, and the phase transitions are signaled by the minimum of the fidelity [23–26]. Here the AE loss is the analogous to the similarity of the input data to the training region learned. The method unveiled the novel region of phase separation between the supersolid and superfluid without invoking the analysis of the order parameter and the energy gaps [27]. However, this method is not fully unsupervised in the sense that brief knowledge of the phase diagram is needed in advance to prepare the training samples. When applying the method on other condensed matter systems, we found that the results depend on the choice of the training region, example can be seen in Appendix B. Moreover, some phase transitions do not show corresponding change with the entanglement spectrum input, necessitating a different input capturing more information and better representation of the quantum state.

In this work, we use the reduced density matrix (RDM) of a many-body system as a better input to the AE to detect phase transitions. With fundamental modifications to the machine architecture, we trained our machine with data expanding over the entire parameter space, thus no prior knowledge of the phase diagram or the order parameter is required. Our scheme successfully identified the rich phase diagram in a variety of one-dimensional models, including the spin-1/2 XXZ model where the transition is of Berezinskii-Kosterlitz-Thouless (BKT) type, the spin-1 XXZ model possessing the topological Haldane phase, and the spinless Su-Schrieffer-Heeger model with interactions. Our approach requires no prior knowledge of the model studied, nor specific training regions, and it works with small amount of training data and on a variety of quantum many-body systems and different types of quantum phase transitions. We also demonstrated the learning ability of the AE by analysing the embedded layer structure of the trained machine and showed that it learns a compressed representation of the states that is distinct for different phases.

## 2 The machine learning model

The methodology employed in this study comprises three main stages and is summarized in the flowchart shown in Fig. 1. In the first stage, data generation is executed using exact

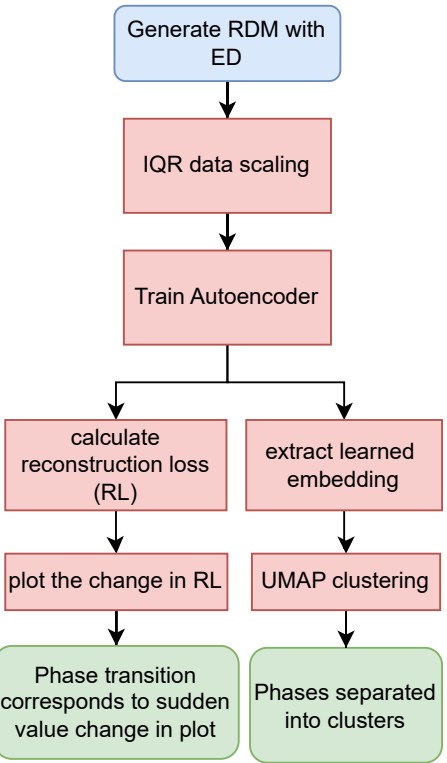

Figure 1: Flowchart illustrating the steps taken to identify quantum phase transitions in a quantum many-body system with a given Hamiltonian.

diagonalization (ED) [28] implemented in the python package QuSpin [29, 30]. While ED simulations are limited to small system size, this is balanced by its ease of implementation and accuracy, making it a valuable technique that is used to study contemporary topics in many-body systems [31, 32]. In addition, ED can simulate a wide range of non-equilibrium and complex systems which can be difficult for other numerical methods. For example, in some non-Hermitian systems, large-sized systems can be prone to numerical instabilities [33] that may hinder the use of iDMRG for the study. The ED method provides us with a numerical solution for the ground state $|\Psi_0(\lambda)\rangle$ of the many-body system at different driving parameters $\lambda$ with high accuracy. The system's half-block RDMs are then calculated by tracing out the degrees of freedom outside the subsystem $A$, i.e. $\rho_A = \text{Tr}_{\notin A} |\Psi_0(\lambda)\rangle \langle\Psi_0(\lambda)|$, and are chosen as the input data since they are rich in information about the system and previous works have shown the capability of using the RDM to derive the potential order parameters of different quantum phases [34, 35]. When simulating the input data, we increment the driving parameter with steps of order 0.01, generating about 200-800 data points for systems with a single driving parameter and 40,000-160,000 data points for systems with two driving parameters. The resulting RDM data is then subjected to a scaling process utilizing the interquartile range (IQR) robust scalar and simple clipping (see Appendix C). This scaling technique is used due to its resilience against outliers, thereby ensuring the data utilized is not skewed.

The second stage involves leveraging the AE, a neural network architecture designed for unsupervised learning. The AE consists of two primary components: an encoder, which maps the input to a lower-dimensional latent representation, and a decoder, which maps this lower-dimensional representation back to the original input space. The dimension of the latent space is usually set to be lower than that of the input to prevent the AE from trivially copying the input to the output. The sparsity of the RDM allows the AE to significantly compress the data while maintaining high accuracy, thus meaningful compressed representations can be learned.

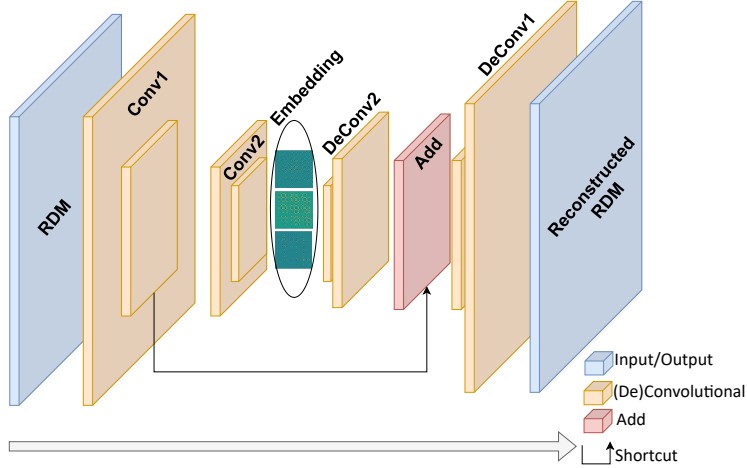

Figure 2: Schematic drawing of the RDM deep learning AE model architecture with shortcut connection.

In this work, we use a deep learning model consisting of two layers of convolutional AE with a shortcut connection across the second layer as sketched in Fig. 2. The symmetric shortcut connection allows information to be passed directly from the encoder to the decoder, bypassing intermediate layers [36]. The feature maps from a shortcut connection and the connected deconvolutional layer are then added, allowing the network to combine information from multiple levels of abstraction. The effect of shortcut connection on the results is discussed in Appendix D.

One application of AE is anomaly detection, where the network is trained on a dataset containing mostly normal or non-anomalous data. The AE encodes and decodes this data, and the reconstruction error - the difference between the original input and the reconstructed output - is calculated. A small reconstruction error indicates that the input data is similar to the training data and is therefore considered as normal. Conversely, a large reconstruction error suggests that the input data differs significantly from the training data and may be anomalous or abnormal. Identifying phase transition is analogous to anomaly detection since data at the transition boundary differs significantly from the data within a phase. Therefore, one may train the AE with data from a single phase and detect the transition from the abrupt increase in the reconstruction error [19]. However, such an approach still requires brief knowledge of the phase diagram to select the training data.

On the other hand, if an AE is trained on multiple distinct types of data, it may exhibit different reconstruction errors for each type. This is because the AE learning rate and compression loss of each data type can differ. Thus, an AE's reconstruction errors may vary for different types of data depending on how well it has learned their respective characteristics during training. This means we can train the AE on entire parameter space containing multiple phases while still being able to distinguish the different phases, achieving the truly unsupervised detection of phase transitions. In this work, we trained our AE across the entirety of the data range for single driving parameter systems and on about 10% of the data chosen randomly for systems with two driving parameters.

Finally, in the third stage, a visualization process is implemented. This is achieved by calculating the Mean Squared Error (MSE), which quantifies the loss between the original input and the AE's reconstruction, i.e.

$$\text{MSE}(A, B) = \frac{1}{n^2} \sum_{i=1}^{n} \sum_{j=1}^{n} \left( A_{ij} - B_{ij} \right)^2 \,, \tag{1}$$

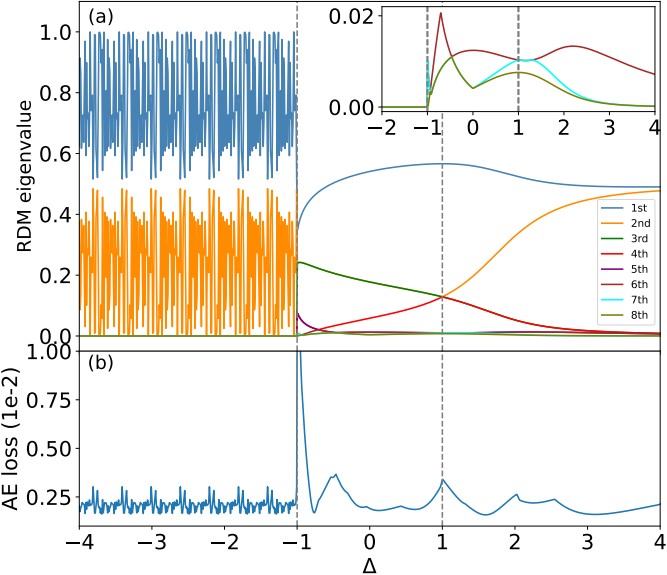

Figure 3: (a) The first eight eigenvalues of the half-block reduced density matrix i.e. the eight largest entanglement spectrum values of the spin-1/2 XXZ chain with $N = 20$ sites. Inset shows zoom-in of the 6th to 8th eigenvalues where spectrum crossing at $\Delta = -0.5$ and local maximum at $\Delta = 2$, which do not correspond to a phase transition, are observed. (b) The AE (trained on entanglement spectrum) loss as a function of the driving parameter. There are peaks at $\Delta$'s not corresponding to the transition points but as a result of the changes in the entanglement spectrum structure. The vertical dashed lines indicate the theoretically predicted critical points.

where $A$ and $B$ is the $n \times n$ input and output matrix respectively. In the following, we will use MSE and AE loss interchangeably but they should be understood as the equation above. The rate of change of the AE loss as a function of the driving parameters is then plotted for analysis. It is postulated that changes in the gradient of this plot can be interpreted as corresponding to a phase transition within the system under study. This is because transition points can act as outliers in data, leading to an abrupt increase in the reconstruction error. Furthermore, different phases will be learned with different accuracy resulting in changes in loss. By observing these changes, we aim to identify the phase transitions in the systems being studied. In addition, we also extracted the learned embedding representation of RDM at the Autoencoder bottleneck and clustered it according to the quantum phases.

# 3 Learning the phase diagrams

We apply the above scheme to several 1D quantum systems, including spin and fermionic models possessing various types of phase transitions. The periodic boundary condition is adopted unless otherwise specified. The results demonstrate the capability of our method in identifying different quantum phase transitions with high accuracy.

## 3.1 Spin-1/2 XXZ model

The Hamiltonian of the XXZ model reads

$$H = \sum_j \left( S_j^x S_{j+1}^x + S_j^y S_{j+1}^y + \Delta S_j^z S_{j+1}^z \right), \tag{2}$$

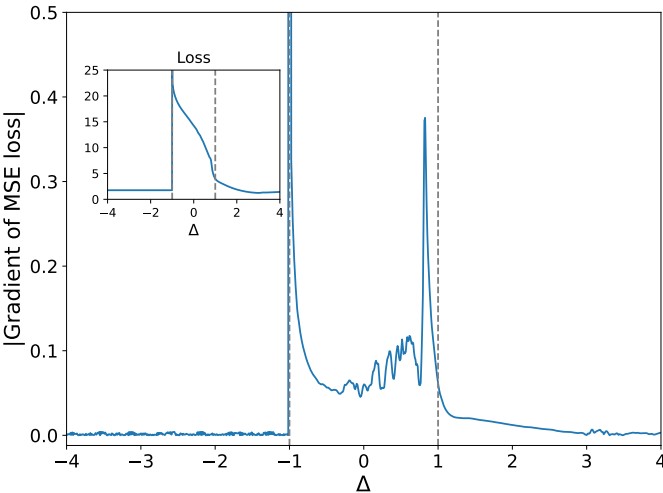

Figure 4: Magnitude of the gradient of the AE loss for the spin-1/2 XXZ model train on half-block RDM. Inset shows the AE loss as a function of $\Delta$. Here $N = 20$. The vertical dashed lines correspond to the theoretical transition points at $\Delta = \pm 1$.

where $S_j^x, S_j^y, S_j^z$ are the spin-1/2 operators and $\Delta$ is the parameter characterising the anisotropy in the spin-spin interaction. The ground state phase diagram consists of three distinct phases: the ferromagnetic (FM) phase, the critical (XY) phase, and the antiferromagnetic (AFM) phase [38–41]. The system experiences quantum phase transitions between these phases with the anisotropy parameter $\Delta$ at -1 and 1 respectively. The XY-AFM transition at $\Delta = 1$ is a Berezinskii-Kosterlitz-Thouless (BKT) type which have been challenging for detection using other methods such as the fidelity susceptibility [25, 42, 43].

In Kottman *et. al.*'s work, they mainly used the entanglement spectrum, i.e. the half-block reduced density matrix eigenvalues, from a single phase as training data for the AE [19]. However, we find that the entanglement spectrum is insufficient and presents some issues when AE is trained on the entire parameter range as shown in figure 3(b). While there are peaks in the AE loss signalling the transitions at $\Delta = -1$ and 1 respectively, there are also other peaks with comparable magnitudes to the one at $\Delta = 1$ (the BKT transition) taking place within a phase at $\Delta = -0.5, 2, 2.5$. This can be understood from the qualitative structure of the entanglement spectrum, where the first eight values as a function of $\Delta$ are plotted in Fig. 3(a). The two peaks at $\Delta = -1$ and 1 reflect the significant changes in the two dominating eigenvalues in the spectrum. However, the lower eigenvalues can also carry non-trivial features, for example, the crossing around $\Delta = -0.5$ and the local maximum around $\Delta = 2$ without the system undergoing a phase transition. This in turn causes the additional peak observed in the loss of the AE trained with the entanglement spectrum.

This shows that using the entanglement spectrum to train on an entire parameter space requires prior knowledge of which eigenvalues to focus on. However, this will then defeat the goal of investigating phase transitions in new models. Even though the transitions in this spin-1/2 XXZ model have significant changes in the dominating eigenvalues, this may not be the case for other models. One example is the Spin-1 XXZ model which we considered in the next section (see Appendix E).

To solve the issue, we turn to training the AE with half-block RDM data from the entire range of $\Delta$. The entanglement spectrum is derived from the eigenvalues of the RDM and offers insights into entanglement of the subsystem and its complement. The RDM, from which the entanglement spectrum is derived, provides a more comprehensive picture, encompassing a complete description of the subsystem's state including the entanglement information. This

Table 1: Comparison of predicted and expected transition points in the spin-1 XXZ model at different values of D. The error in the prediction is taken as the standard deviation, denoted as "std". The expected value is taken from Fig. 1 in Ref. [37] with reading accuracy of 0.1 and an estimated error of ±0.05.

| D = 0 | | |
|---|---|---|
| **Transition** | **Mean prediction ± std** | **Expected** |
| FM-XY | -1.01 ± 0.00 | -1.00 ± 0.05 |
| XY-Haldane | 0.05 ± 0.04 | 0.00 ± 0.05 |
| Haldane-Neel | 1.10 ± 0.03 | 1.20 ± 0.05 |
| D = 0.8 | | |
| **Transition** | **Mean prediction ± std** | **Expected** |
| FM-XY | -1.45 ± 0.00 | -1.50 ± 0.05 |
| XY-LargeD | -0.67 ± 0.05 | -0.60 ± 0.05 |
| LargeD-Haldane | 0.67 ± 0.08 | 0.70 ± 0.05 |
| Haldane-Neel | 1.54 ± 0.03 | 1.70 ± 0.05 |

depth of information within the RDM can make discerning quantum phases difficult. However, with neural networks capabilities analyzing complex data, it becomes feasible to use RDM to identify phase transitions. As such, in our use of AE to detect phase transitions in quantum many-body systems, the RDM stands out as a more advantageous input compared to the entanglement spectrum. Despite the increased input data dimension using RDM as compared to the entanglement spectrum, the training time is within manageable limits using common computational workstations nowadays. Further optimization is also possible by writing custom convolution layers that work on sparse matrix format.

Figure 4 shows the resultant AE loss and its gradient magnitude as a function of $\Delta$, with AE bottleneck size of $128 \times 128$ and RDM size $1024 \times 1024$. There are three main distinctive regions corresponding to the three phases, and the transitions are captured by the abrupt changes in the loss gradient near $\Delta = -1$ and 1. The FM and AFM phases have low AE loss and gradient. On the other hand, the XY phase starts with the highest loss but decreases in a linear like fashion with small fluctuations that plateaus at the XY-AFM transition point. This suggests learning the FM and AFM phases is easier than the XY phase, which is also consistent with the expectation that the XY phase has a more complex order parameter. Note that despite the similarity in concept between the AE approach and the fidelity approach, we managed to detect the XY-AFM transition with GS data while the latter approach needed 1st excited state to detect the transition [26].

## 3.2 Spin-1 XXZ model

We next consider a system with more than one driving parameter. The one-dimensional Spin-1 XXZ Model with uniaxial single-ion-type anisotropy given by the Hamiltonian [37]

$$H = \sum_j \left( S_j^x S_{j+1}^x + S_j^y S_{j+1}^y + \Delta S_j^z S_{j+1}^z \right) + D \sum_j \left( S_j^z \right)^2 , \tag{3}$$

where $S_j^x, S_j^y, S_j^z$ are the spin-1 operators at site $j$, $\Delta$ is the spin-spin interaction anisotropy parameter, and $D$ characterises the uniaxial sinlge-ion anisotropy. The system has a rich ground state phase diagram consisting of a topological Haldane phase, large-D, Neel, FM and XY phases. The system undergoes quantum phase transitions between these phases as the anisotropy parameters $\Delta$ and $D$ are varied.

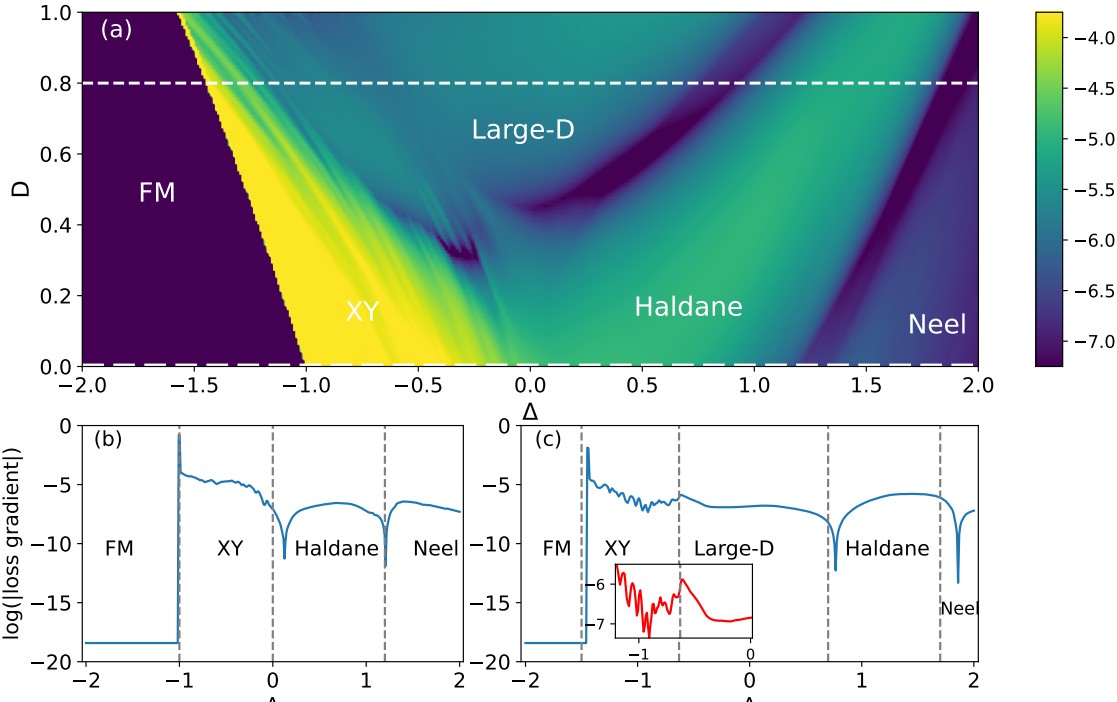

Figure 5: (a) The color map of the logarithm of the magnitude of AE loss gradient for the spin-1 XXZ model with $\Delta$ and $D$ as the driving parameters. The white dashed lines indicate the path of the 1D phase diagram plotted in (b) and (c). (b) and (c) shows the logarithm of the magnitude of AE loss gradient at fixed $D = 0$ and $D = 0.8$, respectively. The inset in (c) focuses on the XY-LargeD transition. The vertical dashed lines correspond to the transition points predicted in [37]. Lattice size of $N = 12$ is used here.

Figure 5(a) shows the magnitude of the AE loss as a function of the driving parameters for the machines trained on RDM data with AE bottleneck size of 81×81 and RDM size 729×729. Given that our training data has increased by over an order of magnitude with two driving parameters, the AE's loss and its gradient are significantly reduced. Therefore, we use a logarithmic scale when plotting to more clearly visualize these changes. Five distinct regions can be identified, among which the regions corresponding to the FM and XY phases are particularly prominent. Although the other regions have a close magnitude of the AE loss, clear boundaries separating these regions can be observed. In figures 5(b) and (c), we extract the logarithmic changes in loss at fixed $D = 0$ and $D = 0.8$ respectively as a function of $\Delta$. The sudden changes in loss gradient align closely with the predicted phase transition points in the literature [37], with minor deviation for Haldane phase transitions which can be caused by difficulty of learning long-range entanglement in Haldane phase and finite size effect could be another factor. This alignment underscores the reliability of the method in estimating the transition points.

To achieve clearer boundaries between the phases, especially between the XY, large-D and the Haldane phases, we trained 50 simple supervised classifier networks (with the architecture presented in Appendix F) on small regions (200 data points) centered within each phase as identified in Fig. 5, and used the networks to predict the phase diagram. The predicted phase diagram with the phase boundary averaged over the 50 runs is shown in Fig. 6, and the transition points for $D = 0$ and $D = 0.8$ are presented in Table 1. The obtained critical points match well with the expected results in Ref. [37] for transitions between the FM, XY, large-

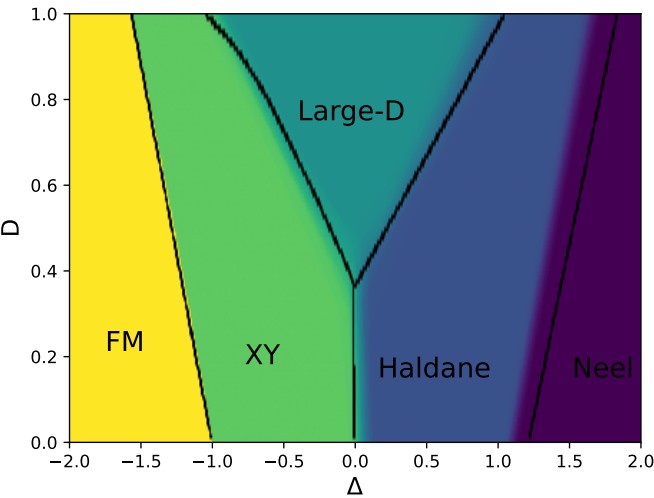

Figure 6: Classifier phase prediction of spin-1 XXZ model. Black lines represents the phase boundaries extracted from Ref. [37].

D, and Haldane phases, with a slight deviation in the Haldane-Neel transition. The slight deviation may attribute to the fact that the topological Haldane phase is more difficult to learn in general due to the long range entanglement and the classifier learned features of the Haldane phase near the Haldane-Neel transition are too similar to learned features of the Neel phase.

After training the AE on the RDM data, it learns a compressed representation in each layer. We examine the learned representation at the bottleneck, where the RDM size has been compressed from $3^6 \times 3^6 = 729 \times 729$ to $81 \times 81$. Figure 7 shows visualizations of the learned representations picked from two points in each phase. The learned representations from each phase show a distinct pattern, indicating the AE's ability to learn distinct features for each phase. However, it is worth noting that this visualization is the output of a channel in the AE's second convolutional layer, i.e. the bottleneck, and only shows the compressed RDM embedding. The emerging distinct patterns cannot be explained because unsupervised neural networks have low explainability and are usually regarded as black boxes.

We further analysed the learned representations by projecting them into 2D feature space using a non-linear dimensionality reduction technique known as Uniform Manifold Approximation and Projection (UMAP) [44]. Dimensionality reduction is a process used in data analysis and machine learning to simplify high-dimensional data into a lower-dimensional form, making it more manageable and computationally efficient. By reducing the number of random variables under consideration, it retains the essential features of the data, thereby facilitating tasks such as data visualization. The UMAP stands out for its effectiveness and efficiency. It operates on the principle that uses Riemannian geometry to construct a graph representation of the high-dimensional data. The algorithm then optimizes a low-dimensional graph to closely resemble the high-dimensional one, resulting in a simplified representation that retains the original data's topological structure. By preserving the global structure of data, UMAP allows for the clear identification of clusters or groups of similar data points, providing valuable insights that are critical in data-driven decision-making processes.

We trained a UMAP transformer on 200 data points from each of the five phases observed in Fig. 5(a) and use it to visualize the learned representations in the AE on a 2D feature space.

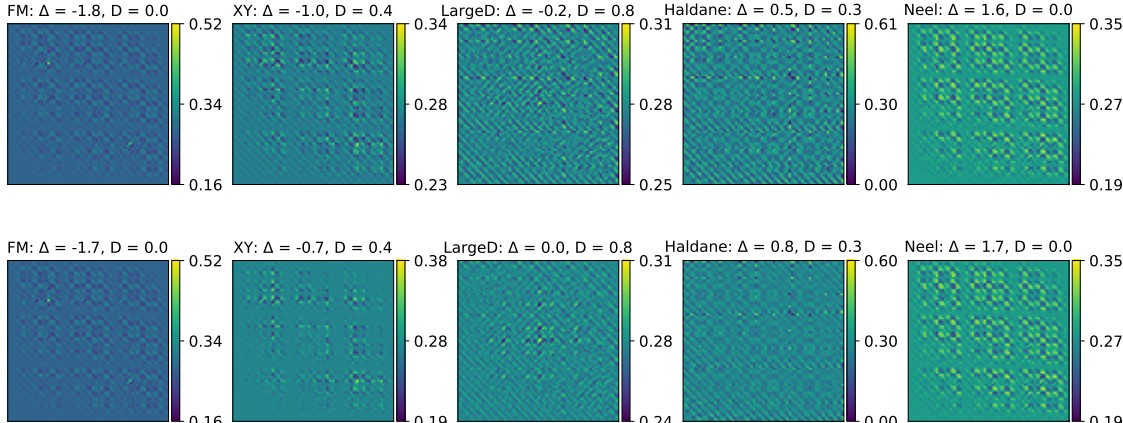

Figure 7: Visualisation of a channel of the second convolutional layer output, i.e. the learned bottleneck embedding with size 81×81, of the five phases in spin-1 XXZ model. The rows show that embeddings are different when $\Delta, D$ are sampled from different phases, while the columns show that the embedding is similar when they are sampled from the same phase.

Figures 8 (a) and (b) show the visualization of the learned representation at $D = 0$ and $0.8$ respectively for $\Delta = [-2, 2]$, which is the same range shown in Figs. 5 (b) and (c), and includes both seen and unseen data. It is clear that each phase data points cluster together forming four separate clusters, with few outliers at transition boundaries, e.g. transition points between XY-Haldane being outliers, this matches with the small deviation of the theoretical transition point at $\Delta = 0$ shown in Figs. 5 (b) and (c). Being able to successfully cluster the phases embedding demonstrates that the learned representation contains information that correlates to the phase properties which makes it potentially useful in future analysis of the phases' order parameters and other machine learning applications [45, 46].

It is worth noting that the phase diagram in Ref. [37] includes a second XY phase 'XY2' phase that is far from the region of interest we studied above. To show that our method is also capable of distinguishing between the XY1 and XY2 phases, we trained an AE with the RDM from $D = -2.5$ to $-1.5$ at fixed $\Delta = -0.1$. The result is plotted in Fig. 9 and we were able to identify the transition point at $D = -1.98$.

## 3.3 The Su-Schrieffer-Heeger model

We further applied the proposed method to a spinless fermion model, namely the Su-Schrieffer-Heeger (SSH) model, which is a foundational model that has been frequently investigated in the study of topological insulators [35, 47]. The interacting SSH model is characterized by the following Hamiltonian:

$$H = -t \sum_j \left[ (1+\eta)c^\dagger_{j,A}c_{j,B} + (1-\eta)c^\dagger_{j,B}c_{j+1,A} + h.c. \right] + U \sum_j n_{j,A}n_{j,B} + V \sum_j n_{j,B}n_{j+1,A}, \quad (4)$$

where $c^\dagger_{j,A(B)}$ and $c_{j,A(B)}$ are the creation and annihilation operators for a spinless fermion at site A(B) in the unit cell $j$, respectively. The parameter $t$ represents the hopping amplitude between the nearest-neighboring sites and is taken to be 1 for convenience, $\eta$ is the parameter characterizing the anisotropy in the intercell and the intracell hopping, and $U$ and $V$ characterize the strength of intracell and intercell interactions respectively, $n_{j,A(B)} = c^\dagger_{j,A(B)}c_{j,A(B)}$ is the number operator at site A(B) of the $j$-th unit cell. In the absence of interactions, the SSH model exhibits a topological phase denoted as $O_-$ for $\eta < 0$ where a quasi-local order

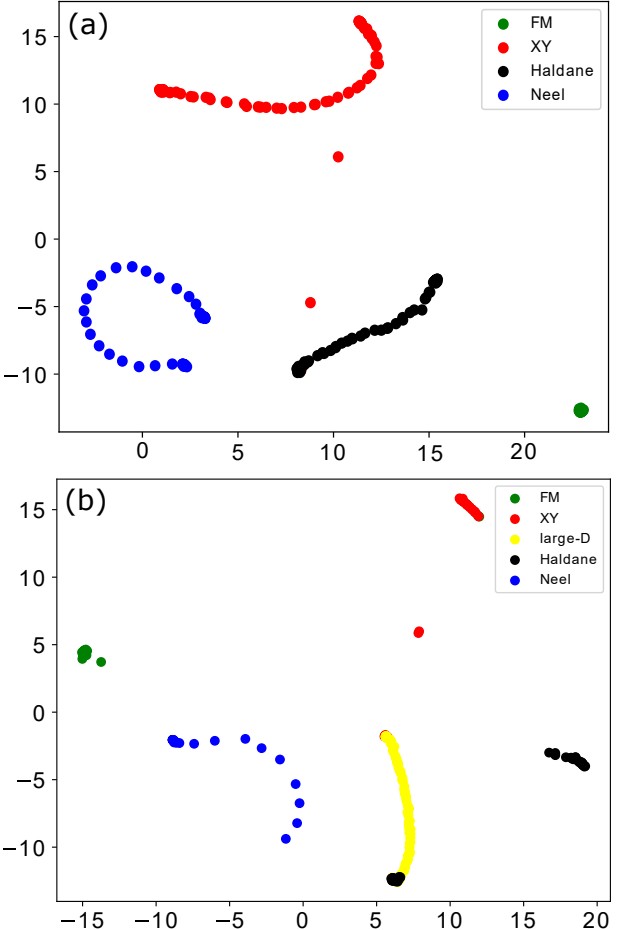

Figure 8: UMAP visualization of the AE learned representation for the spin-1 XXZ model RDM projected onto a 2D feature space. Here the lattice size is $N = 12$, $D = 0$ and 0.8 in (a) and (b) respectively, with $\Delta = [-2, 2]$. The data points are colored with respect to the expected transition points in Ref. [37].

parameter has been identified by careful analysis of the RDM spectrum [35,47]. A topological phase transition takes place at $\eta = 0$ and the system transforms to a trivial phase denoted as $O_+$ for $\eta > 0$ [48]. In the presence of interactions, the model exhibits a rich ground state phase diagram consisting of multiple phases [35].

We study the model at $\eta = -0.6$ and the interaction range $U \in [1.0, 5.0]$ and $V \in [-4.0, 0]$. Such a driving parameters range is chosen to cover most of the phases, namely the topological phase $O_-$, the trivial phase $O_+$, and a charge density wave (CDW) phase, in the model while keeping the training/testing dataset within a manageable size. The logarithm of the magnitude of the AE loss gradient is shown in Fig. 10 with AE bottleneck size of $128 \times 128$ and RDM size $1024 \times 1024$. Two boundary lines representing the transitions between the three phases are clearly observed in plot (a). These transition lines mostly agree with that found in previous works [35,47]. In Fig. 10(b), we plot the logarithmic change of the loss at fixed $U = 3$ as a function of $V$. Sharp spikes are observed at values of $V$ that are consistent with the transition points found in [35]. Observing the loss in Fig. 11, the 'valley' spike at CDW/$O_+$ transiton corresponds to a local minimum in loss between the gradual change of CDW and $O_+$ losses while the 'peak' spike at $O_+$/$O_-$ transition corresponds to abrupt discontinuity between the $O_+$ and $O_-$ losses.

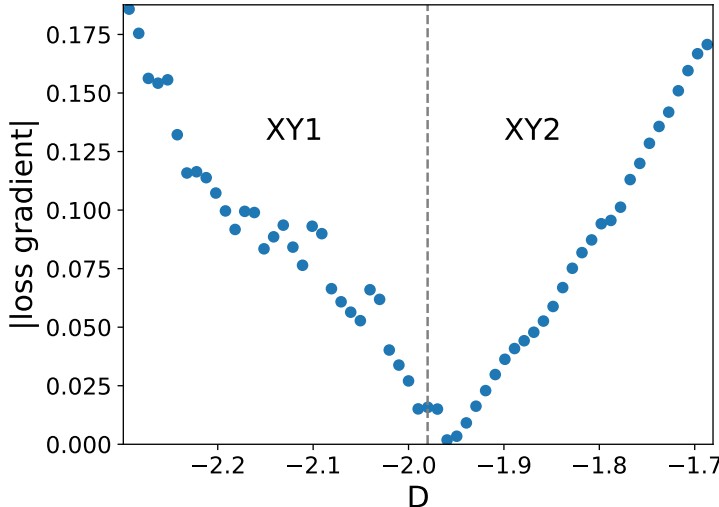

Figure 9: AE loss gradient of spin-1 XXZ model at $\Delta = -0.1$. Dashed line at $D = -1.98$ shows the transition found in Ref. [37] between the XY1 and XY2 phases.

We also study the model at $\eta = -0.6$ and the interaction range $U \in [-1.0, 1.0]$ and $V \in [-4.0, -2.0]$ where the topological phase $O_-$, the trivial phase $O_+$, and the phase seperation (PS) phase reside in. Figure 12 (a) shows the logarithm of the AE loss gradient, where the detected phase boundaries closely match those of Ref. [35,47]. Figure 12(b) plots the logarithmic loss of (a) at $V = -4$, the transition is clearly seen at the expected value of $U = -0.25$. The results further demonstrate the generalizability of our method to identify the phase transition in many-body systems.

## 4 Conclusion

In this work, we have presented an approach for identifying and visualizing quantum phase transitions with minimal prior knowledge using unsupervised machine learning techniques that does not require labeled data and does not need specific regions to train on. Our method is based on neural networks, which enable us to measure changes in the reduced density matrix with driving parameters by analysing the reconstruction loss. We have demonstrated the capability of our method in detecting various types of phase transitions, including topological and BKT transitions, in several quantum systems. No prior knowledge of the order parameters or the phase diagram is required in the process, and our method does not necessitate a large amount of training data and is effective even with small system sizes. This makes the method readily applicable for studying phase transitions in a wide range of novel quantum systems, thus serves as a new tool that complements existing methods by providing new perspectives and broadening the range of the quantum systems that can be explored.

In addition, we showed that relevant features of a phase can be extracted from the compressed representation embedding of the Autoencoder, which can be clustered according to the system's phase with dimensionality reduction techniques such as UMAP. This suggests how quantum states are represented within neural networks and can be useful for further analysis to extract insights into the underlying physics of each phase and may help identify the order parameters.

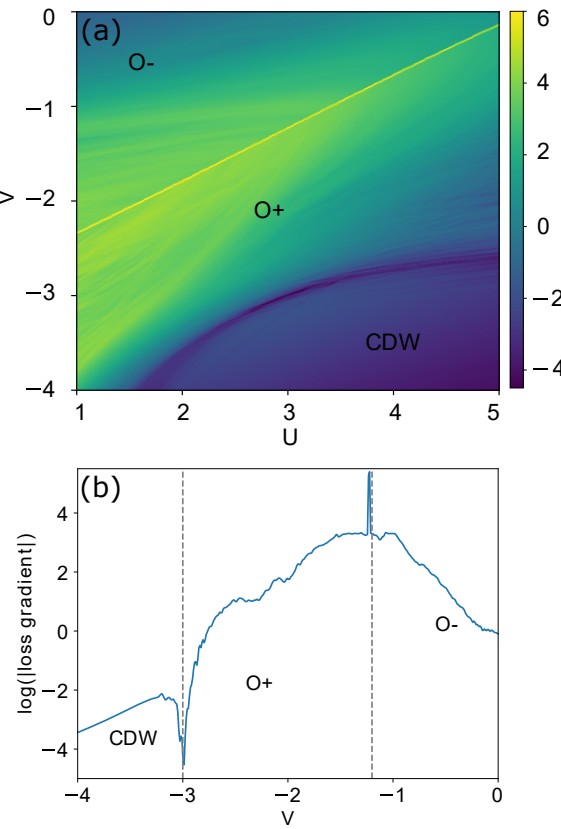

Figure 10: (a) Color map of the logarithm of the absolute AE loss gradient for the interacting SSH model with $U$ and $V$ as the driving parameters. (b) The logarithm of the absolute AE loss gradient as a function of $V$ at fixed $U = 3$. Dashed lines at $V = -3, -1.2$ are the transition points obtained in Ref. [35]. Here $\eta = -0.6$ and a system size of 10 unit cells is considered.

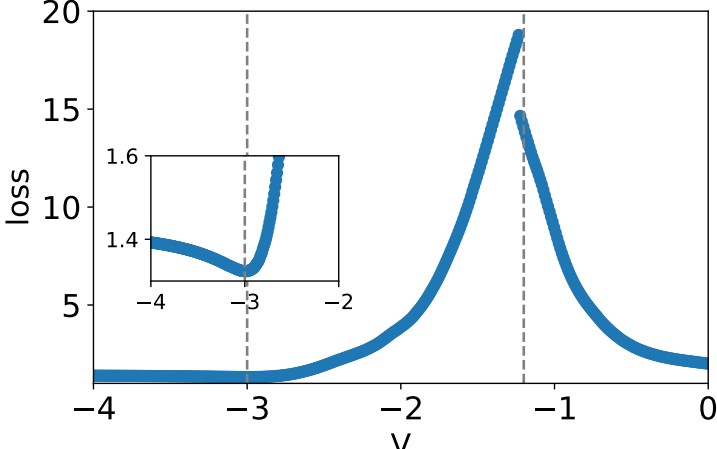

Figure 11: AE loss of the SSH model at $U = 3$. The inset shows the loss magnified at $V = -3$. The CDW/$O_+$ transition at $V = -3$ is a gradual transition with minimum loss at the transition point, while the $O_+/O_-$ transition at $V = -1.2$ is an abrupt discontinuity.

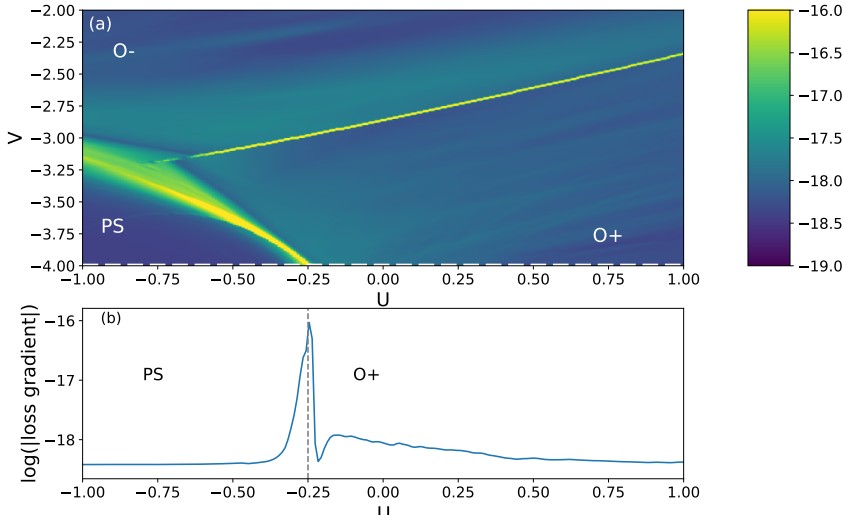

Figure 12: (a) Color map of the logarithm of the absolute AE loss gradient for the interacting SSH model with $U$ and $V$ as the driving parameters. (b) The logarithm of the absolute AE loss gradient as a function of $U$ at fixed $V = -4$. Dashed line at $U = -0.25$ is the transition points obtained in Ref. [35]. Here $\eta = -0.6$ and a system size of 10 unit cells is considered.

Looking forward, the approach described here can be further refined and expanded to tackle even more complex systems. For example, non-equilibrium systems such as the many-body systems with disorders, periodically driven systems with non-equilibrium phases such as discrete time crystals [49, 50]. In addition, we can explore modifying the approach to work with RDM for 6-10 sites blocks from DMRG simulations to overcome the current ED size limitation. The investigation of the performance of the AE in relation, if any, to the nature of the phases and the type of phase transitions will also be an interesting future work.

# Acknowledgments

**Funding information**   We acknowledge financial support from Research Grants Council of Hong Kong (Grant No. CityU 11318722), National Natural Science Foundation of China (Grant No. 12204130), Shenzhen Start-Up Research Funds (Grant No. HA11409065), City University of Hong Kong (Grant No. 9610438, 7005610, 9680320), and HITSZ Start-Up Funds (Grant No. X2022000).

# A   iDMRG correlation length in the spin-1/2 XXZ model

Figure 13 compares the change in the log gradient of AE loss with the correlation length obtained from iDMRG for spin-1 XXZ model in Eq. (3). Both methods accurately detect the topological to non-topological Haldane-Neel transition at $\Delta = 1.2$. However, the XY-Haldane transition, which is believed to be a BKT type, is not detected by the correlation length at $\Delta = 0$. In contrast, our method is able to capture this transition. This demonstrates an example of model specificity for iDMRG correlation length application and our method's potential for broader applicability across different quantum systems.

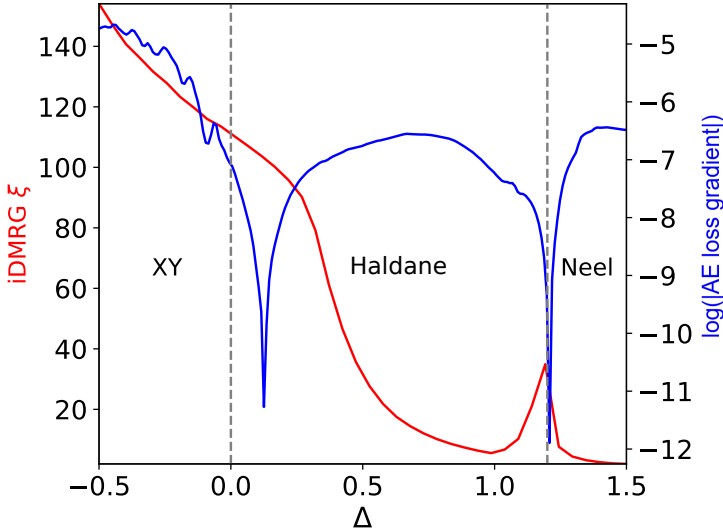

Figure 13: The log gradient of AE loss (in blue) and iDMRG correlation length $\xi$ (in red) as a function of the spin-spin interaction anisotropy parameter $\Delta$ in Spin-1 XXZ model. The uniaxial single-ion anisotropy is set to $D = 0$. A system of 12 sites is considered in the ED simulation in the AE approach. The theoretical phase transitions are indicated by the vertical dashed lines.

## B  Effect of the training region

Figure 14 shows the log(|loss gradient|) results of five AEs, each trained on a 400-point 2D region of a single phase in the phase diagram as specified in Table 2 Depending on the phase chosen, the identified phase transition points can differ; this is especially apparent for the XY-LargeD and LargeD-Haldane transitions. Furthermore, the results from the AE trained on XY phase show additional transitions in the LargeD phase. In Ref. [19],the authors trained the AE on single phase of the entanglement spectrum. Not only does this require prior knowledge of some region of the phase diagram but we have observed that the results produced depend on which phase was chosen for the training.

## C  Data preprocessing

Data preprocessing is a crucial step in preparing data for machine learning algorithms and can significantly impact the performance of the model. One essential aspect of data preprocessing is data scaling, which involves normalizing data to a common range to prevent variables with large ranges from dominating the model. While standard scaling techniques such as z-score

Table 2: Training ranges for Fig. 14.

| Phase | $\Delta$ Range | $D$ Range |
|---|---|---|
| Neel | 1.6 : 1.8 | 0.0 : 0.1 |
| Haldane | 0.5 : 1.0 | 0.35 : 0.55 |
| XY | -1.0 : -0.5 | 0.4 : 0.6 |
| LargeD | -0.25 : 0.25 | 0.75 : 0.85 |
| FM | -1.8 : -1.6 | 0.0 : 0.1 |

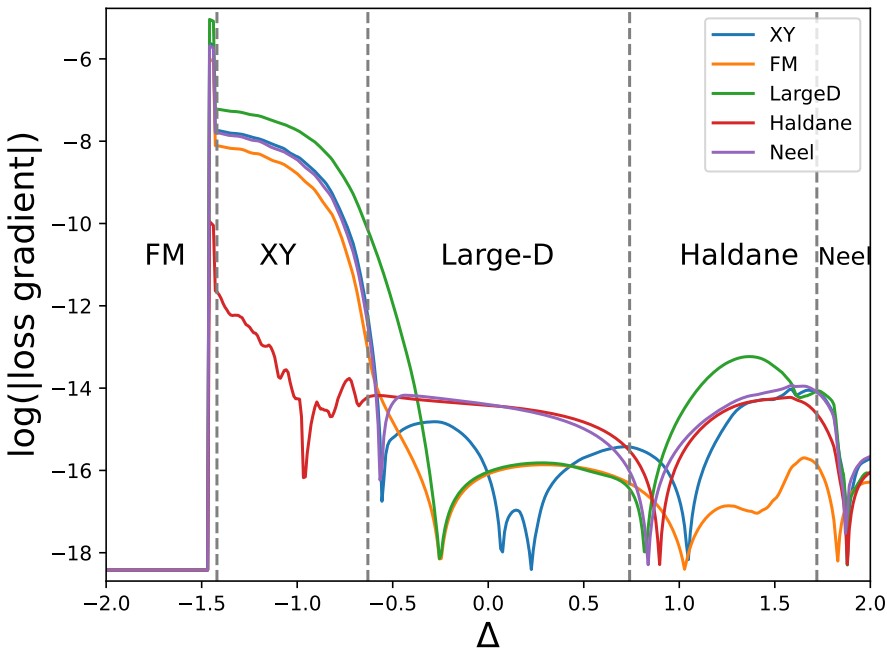

Figure 14: The log(|loss gradient|) of the AE when trained on each of the five regions of the spin-1 XXZ model. Dashed lines are the expected phase transitions.

normalization and min-max scaling are commonly used, they may not be robust to outliers, which can significantly impact model performance and can change data spread (min-max scaling compress data inliers into a narrow range) and data distribution (standard scalar assumes normal distribution of data). In contrast, interquartile range (IQR) robust scaling, as defined by

$$X_{\text{scaled}} = \frac{X - \text{median}(X)}{\text{IQR}(X)}, \tag{C.1}$$

is a technique that can be used to normalize data in the presence of outliers. IQR robust scaling is based on the interquartile range of the data, which is less sensitive to outliers than the mean or standard deviation. We found that IQR followed by clipping outlier values is the best performing scaling technique because the magnitude of the RDM values differs significantly across different phases and we need to scale the data such that all data is of similar order of magnitude. After the IQR robust scaling, we implement additional simple clip scaling to the 99-99.9 percentile of the data to further reduce the influence of outlier points.

# D Effect of shortcut connection

The shortcut used to across the second convolutional layer can sometimes be used to copy results and bypass the bottleneck. However, when compared to AE without the second convolutional and shortcut, the detected transition points as shown in Fig. 15 are less accurate for the AE with shortcut connection. Furthermore, the UMAP clustering in Fig. 8 is implemented on the embedding obtained from the second convolutional layer, which demonstrates that physically meaningful patterns are learned.

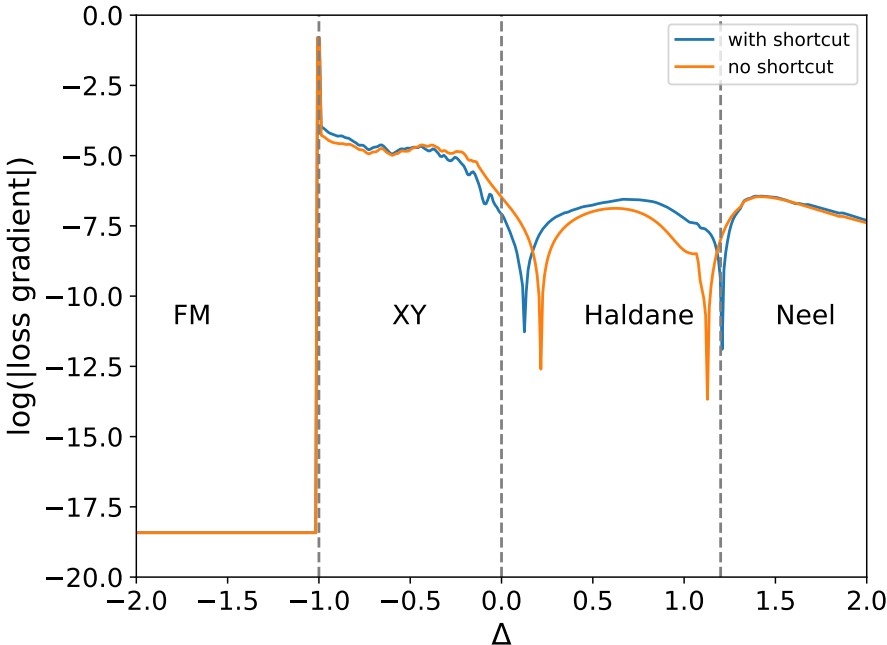

Figure 15: Logarithm of the magnitude of the loss gradient for AE with and without the second convolutional layer and the shortcut connection. Here $D = 0$ and $N = 12$. The vertical dashed lines correspond to the transition points predicted in Ref. [37].

# E   Entanglement spectrum in the spin-1 XXZ model

Figures 16 (a) and (b) show the first few values of the entanglement spectrum of the Spin-1 XXZ. The first three dominating values of the spectrum are featureless at the transition between XY-Haldane phases at $\Delta = 0$. The transition can be only observed starting from the 4th eigenvalue where there is a level crossing. However, besides the features observed at the true critical points, lower eigenvalues also show non-trivial changes at $\Delta = -0.8$ and $\Delta = 1$ which do not correspond to a phase transition. This will in turn lead to extra peaks in AE loss when it is trained with entanglement spectrum data as shown in figure 16 (c). This further demonstrates the deficiency of using ES as input for the AE.

# F   Classifier for phase prediction of the spin-1 XXZ model

Figure 17 and Table 3 shows the schematic drawing and the detailed architecture of the network used for phase classification in the spin-1 XXZ model. The architecture is composed of two convolutional layers with strides and kernel size $3 \times 3$ followed by flatten and two dense layers with final softmax prediction layer that gives probability of each phase. Dropout of 20% is used between dense layers to prevent overfitting.

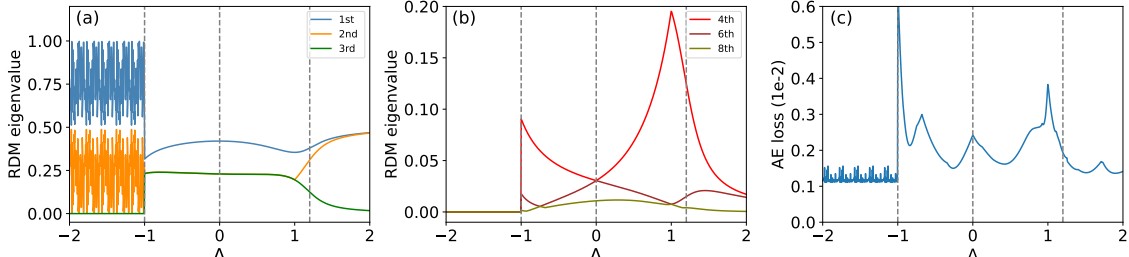

Figure 16: (a) The largest three eigen-values of the half-block RDM of the spin-1 XXZ model of lattice size $N = 12, D = 0$. The values change significantly near the FM-XY transition and the Haldane-Neel transition at $\Delta = -1$ and 1.2 respectively, but not the XY-Haldane transition point at $\Delta = 0$. (b) Tthe 4th, 6th, 8th values of the entanglement spectrum. They show significant features at $\Delta = -0.8$ and 1 which do not correspond to any phase transitions. (c) Loss of AE trained on the entanglement spectrum. Vertical dashed lines indicate the theoretical transition points.

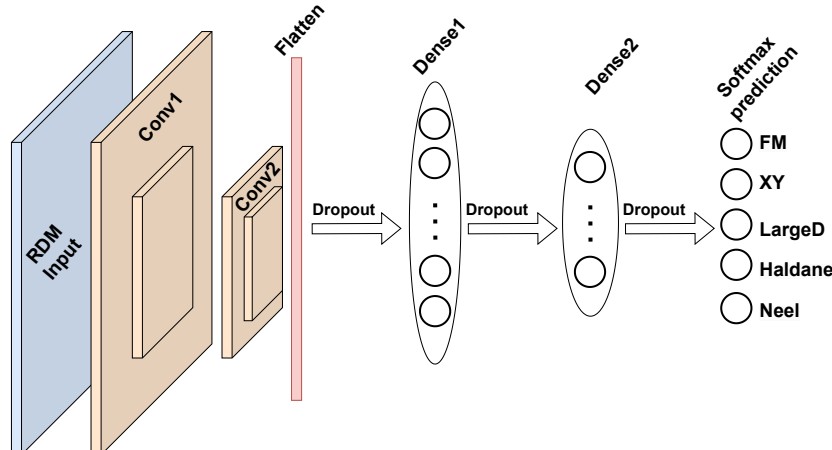

Figure 17: Schematic drawing of the RDM convolutional classifier model architecture.

Table 3: Classifier model architecture.

| Layer Number | Layer Type |
|---|---|
| 1 | Input |
| 2 | Conv2D + pooling |
| 3 | Conv2D + pooling |
| 4 | Flatten |
| 5 | Dropout |
| 6 | Dense |
| 7 | Dropout |
| 8 | Dense |
| 9 | Dropout |
| 10 | Dense (Softmax) |

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
