# Peer review of "Identifying Quantum Phase Transitions with Minimal Prior Knowledge by Unsupervised Learning"

_SciPost Physics Core, doi:SciPost Phys. Core 8, 029 (2025)_

## Round 1 · Referee Report · Anonymous (Referee 1) · 2024-10-11

Report

The article is a correct extension of Kottmann et al. ([19] in the manuscript). Results are presented in a clear manner, and bibliography on previous work seems to be addressed properly. If the remarks bellow are correctly address by the authors I recommend the publication of this manuscript in SciPost Core instead of SciPost Physics, since the results do not "provide details on groundbreaking results obtained in any (sub)specialization of the field."
Limitations of the method proposed are not properly addressed.

Remarks:

1. In Fig. 3, caption and corresponding text, could you emphasize more that you are working with the entanglement spectrum input?

2. In page 7 the authors claim

" There are three main distinctive regions corresponding to the three phases, and the transitions are captured by the abrupt changes in the loss gradient near ∆ = −1 and 1."

But in Fig. 4 the transition is smooth at \Delta = + 1. This seems to happen also in Fig. 3 inset a) of [19]. So in both cases the transition is not sharply signaled. Even if this transition is typically difficult to see, actually your Fig. 3 has a clear cusp at \Delta = + 1, so apparently the combination of your method of training and the entanglement spectrum as input actually also helps validating the results using the whole RDM as input. I think a comment on this can be made.

3. When changing from entanglement spectrum as input to the reduced density matrix you are enlarging the size from n eigenvalues to n**2 matrix entries. This is a drawback of your approach and it should be mentioned in the manuscript. Any possibility (or absence of it) of taking advantage of sparsity in writing the RDM as input should be mentioned.

4. Results on Fig. 9 are very close to the results of Ref. [35], but there are larger discrepancies seen in the cases of Fig. 5 and Fig. 6 and Ref [37]. These discrepancies are not well addressed and some clarifications on the level of precision of Ref. 37 should be made if it is considered as ground truth. Is there any way of improving your results or reducing the discrepancies? (Enlarging system size, changing the architecture, enlarging the network, adding more points for the Haldane phase, etc?)

5. Any comment on why Fig. 9 b) presents a peak or a valley for both transitions?

6. In Fig. 9.a U is in [1,5], whereas in Ref. [35] their phase diagram (fig. 7b) has U in [-5,5]. Their "PS" area is left aside in your manuscript without any comment on it. Please justify properly why you left it aside or add it for completeness.

7. Same as last comment applies for your Fig. 6 and the phase diagram of the S=1 XXZ model in Ref. [37], where the region with negative \Delta even displays another phase called "XY2". Please justify properly why you don't study this region or add it for completeness.

8. At the end in the conclusions you claim that you could study higher dimensional systems. Nonetheless if you need the reduced density matrix as input, that means that you cannot go to any reasonable size to study a 2D or 3D system, right? Even in the present manuscript the system lengths are extremely short, which is a mayor problem of your approach.
Please justify on the algorithmic scalability of your setup to seriously consider this perspective.

Best regards.

Recommendation

Accept in alternative Journal (see Report)

---

## Round 1 · Referee Report · Anonymous (Referee 2) · 2024-11-11

Report

The use of ML methods to identify (new) phases of matter has been a popular subject for several years, however, most studies suppose a prior knowledge of the phases diagram of the system. This article proposes a possible new method that circumvents this issue and that could at term, allow the discovery of new phases.

1. Line 79: the authors claim "We found that the results depend on the choice of the training region."
In this sense, it would have been interesting to have more details on the regions trained, maybe by marking them on the phase diagram.

2. Line 120-121: "The dimension of the latent space is usually set to be lower than of the input to prevent the AE from trivially copying the input to the output".
Could you please give the exact dimensions of the input and the latent space? How does the size of the input affect the outcome of your training? Furthermore, the authors use an AE architecture with a skip connection going from the encoder to the decoder, wouldn't this connection contribute to copying? It would be interesting to see how the feature maps/weights of the decoder look before and after the added skipped layer (it might not be necessary for the main article but simply as a sanity check, that could appear in the annex).

3. "There are three main distinctive regions corresponding to the three phases, and the transitions are captured by the abrupt changes in the loss gradient near $\Delta=-1$ and $\Delta=1$."
In fig. 4 there is a clear abrupt change in the gradient of the loss at $\Delta=-1$, however, the second change in the gradient occurs clearly before the expected transition at $\Delta=1$. The authors fail to explain this behavior.

4. Fig. 5: What are the horizontal dashed lines in a)? Please explain it in the caption. Furthermore, the phase diagram presented in ref. [37] fig 1. spans from D=-4 to D=4. Why did the authors reduce the region trained?

5. Line 225-227: "Five distinct regions can be identified, among which the regions corresponding to the FM and XY phases are particularly prominent. Although the other regions have a close magnitude of the AE loss, clear boundaries separating these regions can be observed."
Looking at fig. 5, a clear change appears close to the expected FM-XY transition point in b) and c), and the same for the Haldane-Neel in fig. 5 b). However, in fig. c) the XY-Large D transition is not signaled by an important change in the gradient. Furthermore, the XY-Haldene in fig b) and Haldane-Neel fig c) are shifted compared to the expected transition point. How do the authors explain this? It seems that despite the claim of success of this method without prior knowledge of the phase diagram, prior knowledge was used to identify the "five distinct regions".

6. Fig. 7: Could the authors explain the patterns that seem to emerge?

7. Fig. 8: Separate clusters corresponding to the different phases appear to exist. From my understanding this is the UMAP visualization on the training dataset, did the authors perform a visualization of an unseen dataset (i.e. test dataset)? Are the different clusters still present?

8. Fig. 9: Again, the phase diagram in ref. [35] spans a larger region, why did the authors choose to focus on this region?

9. Fig. 9 proves to be more convincing than fig. 4, 5b) and 5c). How can the authors explain that their methods appear to work better on SSH model? Can we conclude on the nature of the models where this method would work better?

Overall, this a clear and well-written article, however, several points need to be addressed before it is published by a journal. Furthermore, despite the ambitious claims, knowledge of the phase diagram appears to be used several times (fig. 4, fig 5 b), fig. 5 c) or for the classification fig. 6). As such, even if this study is a step in the right direction it is unfortunately not yet groundbreaking. As such I would recommend a publication in another journal.

Recommendation

Accept in alternative Journal (see Report)

---

## Round 2 · Referee Report · Anonymous (Referee 1) · 2025-1-19

Report

The authors have replied to my comments, nonetheless I have a remark to make regarding R2.

Fig. 3.b) shows that the system, at the given N=20 size, has the transition at Delta = +1, so when you don't see the transition here, it's doesn't seem to be a finite size effect of the system. There could be a finite size effect of your AE. You showed in the new Fig. 15 that your transition points are strongly affected by the shortcut (which is quite surprising to me). Then I think it's expected for your transition points to improve if you make your network larger and better performing. This should be emphasized.

Recommendation

Publish (meets expectations and criteria for this Journal)

---

## Round 2 · Referee Report · Anonymous (Referee 2) · 2025-2-6

Report

The authors responded to all the points raised in my previous report. Furthermore, I am satisfied with the changes in the main text and the appendix.

Recommendation

Publish (meets expectations and criteria for this Journal)

---

## Round 2 · Author Response

List of changes

1. Lines 124-125, added comments about RDM sparsity effect in learning compressed embedding.
2. Fig. 3 caption, added clarification that entanglement spectrum is used as input
3. Fig.4 , replace old graph with corrected new one
4. Lines 208-211, addressed the disadvantage of increased input dimension due to changing input from ES to RDM
5. Lines 213,232,314 : specified the input and embedding dimensions of the data.
6. Fig. 5 c, added inset showing the XY-LargeD transition
7. Lines 241-243 : added possible explanation of Haldane phase transitions result deviation from reference.
8. Lines 262-264, Fig. 7 caption , added explanation of the visualization seen in Fig. 7
9. Added Fig. 9 to show XY1-XY2 transition.
10. Added Fig.11 to show SSH loss and explain the ‘peak’, ‘valley’ in Fig. 10
11. Added Fig.12 to show PS phase transitions.
12. Modified the conclusion, replacing studying higher dimensions system in future works to modifying method to work on block RDM obtained from DMRG simulations
13. Added appendix B to show effect of training region
14. Added appendix D to show effect of shortcut connection

---

## Round 2 · List of Changes

1. Lines 124-125, added comments about RDM sparsity effect in learning compressed embedding.
2. Fig. 3 caption, added clarification that entanglement spectrum is used as input
3. Fig.4 , replace old graph with corrected new one
4. Lines 208-211, addressed the disadvantage of increased input dimension due to changing input from ES to RDM
5. Lines 213,232,314 : specified the input and embedding dimensions of the data.
6. Fig. 5 c, added inset showing the XY-LargeD transition
7. Lines 241-243 : added possible explanation of Haldane phase transitions result deviation from reference.
8. Lines 262-264, Fig. 7 caption , added explanation of the visualization seen in Fig. 7
9. Added Fig. 9 to show XY1-XY2 transition.
10. Added Fig.11 to show SSH loss and explain the ‘peak’, ‘valley’ in Fig. 10
11. Added Fig.12 to show PS phase transitions.
12. Modified the conclusion, replacing studying higher dimensions system in future works to modifying method to work on block RDM obtained from DMRG simulations
13. Added appendix B to show effect of training region
14. Added appendix D to show effect of shortcut connection

---

## Editorial Decision

published